# Liquid-Phase Dehydration of Glycerol to Acrolein with ZSM-5-Based Catalysts in the Presence of a Dispersing Agent

**DOI:** 10.3390/molecules28083316

**Published:** 2023-04-08

**Authors:** Lin Huang, Bo Wang, Licheng Liu, Armando Borgna

**Affiliations:** Institute of Sustainability for Chemicals, Energy and Environment, Agency for Science, Technology and Research, 1 Pesek Road, Jurong Island, Singapore 627833, Singapore

**Keywords:** liquid-phase, dehydration of glycerol, acrolein, sulfolane, ZSM-5, H_3_PO_4_-modified, heteropolyacid, Brønsted acid sites

## Abstract

Liquid-phase dehydration of glycerol to acrolein was investigated with solid acid catalysts, including H-ZSM-5, H_3_PO_4_-modified H-ZSM-5, H_3_PW_12_O_40_·14H_2_O and Cs_2.5_H_0.5_PW_12_O_40_, in the presence of sulfolane ((CH_2_)_4_SO_2_) as a dispersing agent under atmospheric pressure N_2_ in a batch reactor. High weak-acidity H-ZSM-5, high temperatures and high-boiling-point sulfolane improved the activity and selectivity for the production of acrolein through suppressing the formation of polymers and coke and promoting the diffusion of glycerol and products. Brønsted acid sites were soundly demonstrated to be responsible for dehydration of glycerol to acrolein by infrared spectroscopy of pyridine adsorption. Brønsted weak acid sites favored the selectivity to acrolein. Combined catalytic and temperature-programmed desorption of ammonia studies revealed that the selectivity to acrolein increased as the weak-acidity increased over the ZSM-5-based catalysts. The ZSM-5-based catalysts produced a higher selectivity to acrolein, while the heteropolyacids resulted in a higher selectivity to polymers and coke.

## 1. Introduction

Due to the limited reserve of fossil fuel, biomass is considered as an alternative feedstock to produce either fuel or value-added chemicals. As a main by-product of biodiesel production, glycerol can be a preferable platform to derive many value-added chemicals due to its high functionality and surplus supply from biodiesel production [1,2]. A variety of valuable chemicals, such as acrolein, ethylene glycol and propanediols, are known to arise from glycerol by catalytic processes, including dehydration, oxidation, reforming, hydrogenolysis, esterification and oligomerization [3,4,5,6]. Dehydration of glycerol to acrolein is one of the most promising routes because acrolein is an important intermediate for the production of acrylic acid, acrylic acid esters, super-absorbent polymers and detergents [3,4,5,6]. 

Catalytic dehydration of glycerol to acrolein is an acid-catalyzed reaction. Potential acid catalysts for liquid- and gas-phase glycerol dehydration reactions include conventional acids [2], zeolites [2,5,6], heteropolyacids [2,5,6], metal phosphates and sulfates [2,5,6], other solid acids [2,5,6] and ionic liquids [5]. Among the reported catalytic processes, gas-phase glycerol dehydration reactions account for the majority of them, while liquid-phase reactions occupy only a small minority. 

Liquid-phase dehydration of glycerol to acrolein was reported first by Groll and Hearne in a patent in 1936 [7]. In a semibatch reactor, dehydration reaction of an aqueous 6.3% glycerol solution was carried out in the presence of 8% H_2_SO_4_ at 190 °C. The product was recovered by distillation with a yield of 49%, corresponding to a turnover number (TON) of 0.57 g_acr_ g_cat_^−1^. A follow-up of the research was reported by Hoyt and Manninen in a patent in 1951 [8]. In a semibatch reactor, dehydration of aqueous 95% glycerol solution was conducted in the presence of diatomaceous earth-supported H_3_PO_4_ and petroleum oil at ca. 286 °C. The product was recovered by distillation with a yield of 72%. In this area, research did not advance until the end of the 20th century when cheaper glycerol from the biodiesel production process became available. Neher et al. reported liquid-phase dehydration of glycerol to acrolein in a fixed-bed reactor with various heterogeneous catalysts, such as γ-Al_2_O_3_, H_3_PO_4_/α-Al_2_O_3_, Li_3_PO_4_/pumice, Li_3_PO_4_/SiO_2_, FePO_4_/SiO_2_, FePO_4_/pumice, Na-zeolite A, H-ZSM-5 and mordenite, in 1994 and 1995 [9,10]. H-ZSM-5 with H_o_ < −8.2 gave rise to glycerol conversion of 19% and selectivity to acrolein of 71% from aqueous 10% glycerol at 300 °C, 70 bar and an LHSV of 2 h^−1^, corresponding to a turnover frequency (TOF) of 0.038 g_acr_ g_cat_^−1^ h^−1^ [10]. The authors claimed that zeolite catalysts with H_o_ < 2 are suitable for liquid-phase dehydration of aqueous glycerol [9,10]. Nearly a decade later, Suzuki and Takahashi reported a series of solid catalysts with −5.6 ≤ H_o_ ≤ 3.3, such as H_3_PO_4_/Al_2_O_3_, MgSO_4_, KHSO_4_, K_2_SO_4_, Al_2_(SO_4_)_2_ and Al_2_O_3_, for liquid-phase dehydration of glycerol using liquid paraffins as a dispersing agent [11]. The reaction was performed in a batch reactor at 280 °C under atmospheric pressure N_2_, with the product being recovered by distillation. The use of this series of solid catalysts together with the liquid paraffins allowed the authors to achieve acrolein yields of 33–80%. This work was followed up by Takanori and Masayuki who reported alternative solid catalysts, such as FePO_4_, CuSO_4_ and K_0.5_H_2.5_PO_4_/SiO_2_, using dispersing agents with water [12,13]. In a batch reactor at 250–320 °C under atmospheric pressure N_2_, acrolein yields of 61–85% were obtained [12]. Afterward, de Oliveira et al. reported comparative catalytic behavior of various solid catalysts, such as mordenite, ZSM-23, SBA-15, HY and Hβ, in liquid-phase dehydration of glycerol at 250 °C under 70 bar N_2_ in an autoclave [14]. Among these catalysts, the activity was found to have the same order as that of acidity: HY > Hβ > mordenite > SBA-15 > ZSM-23. The pore blockage by heavy polycondensed and cyclic C_6_ compounds resulting from the reaction was assumed to be the main cause of catalyst deactivation. In agreement with such effects of acidity and pore size of solid catalysts on the activity of solid catalysts, Estevez et al. reported that the activity of their solid catalysts for liquid-phase dehydration of glycerol follows the order of AlPO_4_ > HY = H-ZSM-5(SiO_2_/Al_2_O_3_ = 30 molar ratio) > H-ZSM-5(SiO_2_/Al_2_O_3_ = 50 molar ratio) at atmospheric pressure, combining both the acidity and pore size of the catalysts [15]. During this period, Shen et al. investigated liquid-phase dehydration of glycerol over heteropolyacids and over Brønsted acidic ionic liquids in a semibatch reactor at 260–320 °C and atmospheric pressure [16,17]. A maximum acrolein yield of 79% was achieved at 300 °C over H_4_SiW_12_O_40_·15H_2_O (HSiW) for heteropolyacid-catalyzed dehydration of glycerol [16]. The activity of the heterpolyacids for the production of acrolein had an order of HSiW > H_3_PW_12_O_40_·14H_2_O (HPW) > H_3_PMo_12_O_40_·28H_2_O, revealing that liquid-phase dehydration of glycerol is affected by the acidity and stability of heteropolyacids [16]. A maximum acrolein yield of 57% was produced at 270 °C over [Bmim]H_2_PO_4_ for Brønsted acidic ionic liquid-catalyzed dehydration of glycerol [17]. The Bmim cation and moderate acidity were shown to favor the production of acrolein in liquid-phase dehydration of glycerol [17].

From the available literature on liquid-phase dehydration of glycerol to acrolein, liquid-phase dehydration possesses certain potential advantages. Unlike in gas-phase dehydration, evaporation of glycerol is not required in liquid-phase dehydration, which is economic from an energy-saving point of view. There is a broader acid strength suitability of solid catalysts for liquid-phase dehydration. For example, according to Neher et al., zeolite catalysts are suitable with H_0_ < 2 for liquid-phase dehydration, whereas zeolite catalysts are suitable only with −8 < H_0_ < 2 for gas-phase dehydration [9,10]. Moreover, the unique use of dispersing agents favors increasing selectivity to acrolein in liquid-phase dehydration by reducing the condensation of glycerol and by-products [8,11].

Table 1 summarizes the catalytic studies in the literature and shows the new catalytic findings in our work regarding liquid-phase dehydration of glycerol. In the present paper, we present more detailed investigations on liquid-phase dehydration of glycerol to acrolein with ZSM-5-based catalysts in the presence of sulfolane as a dispersing agent under atmospheric pressure N_2_ in a batch reactor, as shown in Figure 1. The choice of ZSM-5 was aimed at providing suitable acid strength and acidity for catalytic dehydration of glycerol to acrolein, as well as bringing about a shape selectivity in favor of production of acrolein. Thus far, only a couple of cases of ZSM-5-based catalysts applied in liquid-phase dehydration of glycerol to acrolein have been reported [10,15]. These investigations were not comprehensive. The reaction was conducted either at high pressures (70–180 bar) [10] or with high acid strength ZSM-5 [10,15]. The use of high acid strength ZSM-5 turned out to either facilitate the formation of condensed glycerol, condensed by-products and coke or cause easy catalyst deactivation [15,18,19,20,21]. The use of dispersing agents in combination with zeolite-based catalysts has yet to be reported. Furthermore, there is a lack of investigation on the relationship between the production of acrolein and the catalyst acid properties in the liquid phase. The lack of such a relationship hinders the catalyst design for efficient liquid-phase dehydration of glycerol to acrolein. Although the dehydration pathway from glycerol to acrolein has long been suggested to be associated with Brønsted acid sites [22], convincing spectroscopic evidence has yet to be provided. In the present work, we investigated lower acid strength ZSM-5 (with higher SiO_2_/Al_2_O_3_ molar ratios) and H_3_PO_4_-modified H-ZSM-5, in combination with sulfolane as the dispersing agent, in liquid-phase dehydration of glycerol to acrolein for the first time. We examined the influences of essential reaction parameters, such as reaction temperature, catalyst/glycerol ratio and sulfolane, on the production of acrolein. We established the relationship between the selectivity to acrolein and the catalyst acidity. In the meantime, the catalyst active acid sites for dehydration of glycerol to acrolein were clearly identified by infrared (IR) spectroscopy of pyridine adsorption. Finally, a high-boiling point solvent, such as sulfolane, proved to be effective for mitigating the formation of polymers and coke.

## 2. Results and Discussion

### 2.1. Influence of Reaction Parameters

The influences of the reaction parameters, including reaction temperature, catalyst/glycerol ratio and sulfolane as the dispersing agent, on the reaction outcomes were first studied with H-ZSM-5 (SiO_2_/Al_2_O_3_ = 280 molar ratio), i.e., ZSM-5_280 as a representative catalyst. H-ZSM-5 with an SiO_2_/Al_2_O_3_ molar ratio of 250–1000 is a known catalyst for gas-phase dehydration of glycerol to acrolein [18,19,23]. In the solvent-free liquid-phase dehydration of glycerol, no reaction occurred in the absence of the catalyst at 200 °C. While at 280 °C, the glycerol conversion and the selectivity to acrolein became 12 and 4.5% after 5 h of reaction, respectively, corresponding to a TON of 0.11 g_acr_ g_cat_^−1^ (Table 2). This indicates that non-catalytic liquid-phase dehydration of glycerol is quite slow, even at high temperatures. Upon the addition of the catalyst with a catalyst/glycerol mass ratio of 0.05, the glycerol conversion and the selectivity to acrolein increased to 16 and 5.8% at 200 °C, respectively, corresponding to a TON of 0.19 g_acr_ g_cat_^−1^. Once the temperature was increased to 280 °C, the glycerol conversion and the selectivity to acrolein remarkably increased, while the coke deposit was reduced from 10 to 3.9%. Both at 200 and 280 °C, di-glycerol and tri-glycerol were observed by GC-MS, clearly indicating the occurrence of glycerol polymerization. The reaction results in Table 2 suggest that high temperatures favor the acrolein yield and suppress the formation of coke in liquid-phase dehydration of glycerol. The coke deposit originates from the polymerization of glycerol and products.

In the solvent-free liquid-phase dehydration of glycerol at 280 °C, the increase in catalyst amount enhanced glycerol conversion and the selectivity to acrolein up to a catalyst/glycerol mass ratio of 0.025 (Table 3). However, it was noted that the reaction rate, i.e., the TOF for glycerol conversion, decreased with increasing catalyst amount, varying from 10 to 1.8 g_gly_ g_cat_^−1^ h^−1^ at catalyst/glycerol mass ratios of 0.005–0.05 (Table 3). This phenomenon is definitely an indication of mass transfer limitations under the reaction conditions [24,25]. As the reaction suffers from higher catalyst/glycerol mass ratios, external and/or internal mass transfer limitations certainly occur [24]. In this circumstance, the TON (or TOF) for the production of acrolein decreased with increasing catalyst amount as well, varying from 5.2 to 1.0 g_acr_ g_cat_^−1^ (or from 1.0 to 0.20 g_acr_ g_cat_^−1^ h^−1^). It follows that higher catalyst/glycerol mass ratios do not favor the production of acrolein from liquid-phase dehydration of glycerol due to the mass transfer limitations. In this respect, decreased TONs (or TOFs) for the production of acrolein with increasing catalyst amount can be likewise observed in the reported work by Estevez et al. on the liquid-phase dehydration of glycerol to acrolein with H-ZSM-5, HY and AlPO_4_ [15]. It is, hence, deduced that mass transfer limitations occur under the reaction conditions with these catalysts as well, in agreement with what has been observed in our case with ZSM-5_280. In the meantime, the catalytic results in Table 2 and Table 3 illustrate that, under the solvent-free conditions, the severe formation of polymers and coke are inevitable due to the predominant inter-molecular dehydration of glycerol over the intra-molecular dehydration of glycerol.

In order to mitigate the polymerization of glycerol and products and improve the selectivity to acrolein, we next used sulfolane as the solvent to disperse glycerol. Sulfolane was chosen because of its high boiling point (285 °C) and high miscibility with glycerol. It can be basically retained in the liquid phase during reaction at below 285 °C. On the other hand, in a liquid-phase reaction, the diffusivity of molecules depends not only on the physical properties of diffusing molecules and the catalyst but also on the nature of the solvent [25]. The choice of solvent is critical for the control of reaction kinetics. The use of sulfolane as the solvent enables appreciable reduction in the viscosity of the liquid-phase reaction system because of the much lower viscosity of sulfolane (10.1 mPa·s at 25 °C [26] than that of glycerol (964 mPa·s at 25 °C [27]). The reduced viscosity was expected to greatly facilitate the diffusion of glycerol and products and, thus, eliminate or minimize the mass transfer limitations. It is known that the diffusivity of a dilute solute in a solvent is inversely correlated with the viscosity of the solvent according to an empirical estimation expression developed by King et al. [25,28].

Figure 1 shows the variations in glycerol conversion and selectivity to acrolein with the reaction time during the liquid-phase dehydration of glycerol over ZSM-5_280 using sulfolane as the solvent at 280 °C and a catalyst/sulfolane/glycerol mass ratio of 0.05/3/1. At initial stage, the inter-molecular dehydration of glycerol dominated, and, thus, the selectivity to high-boiling-point products was predominant. As the reaction proceeded, the intra-molecular dehydration of glycerol speeded up, and, thus, the selectivity to acrolein rose at the expense of the selectivity to high-boiling-point products. The high-boiling-point products mostly comprise of di-glycerol and tri-glycerol. In our reaction system, glycerol, di-glycerol and tri-glycerol are supposed to stay in chemical equilibria. It is deemed that the equilibria can favorably shift toward glycerol in the presence of the dispersing agent. Optimal catalytic results were observed after 5 h of reaction. On this basis, Table 4 shows the effect of sulfolane/glycerol ratio on the liquid-phase dehydration of glycerol over ZSM-5_280 at 280 °C and a catalyst/glycerol mass ratio of 0.05. In a blank test, there was no organic product detected in the presence of ZSM-5_280 and sulfolane without glycerol at 280 °C, indicating that sulfolane is probably stable under the reaction conditions studied here. As the sulfolane/glycerol mass ratio increased, the glycerol conversion and the selectivity to acrolein significantly increased and then decreased. The reaction rate and the TON for the production of acrolein followed the same variation. An optimal selectivity to acrolein of 46% was obtained with complete glycerol conversion at a sulfolane/glycerol mass ratio of 3, corresponding to an optimal TON of 9.2 g_acr_ g_cat_^−1^. Meanwhile, it was found that the selectivity to high-boiling-point products was significantly reduced (from 83 to 47%) when sulfolane was added. It is noteworthy that the TON for the production of acrolein could be increased by 8.2 times, as well as the reaction rate could be enhanced by more than 1.2 time, by the use of sulfolane in the reaction. The reduction in the selectivity to high-boiling-point products in favor of a rise in the selectivity to acrolein implies that the inter-molecular dehydration of glycerol can be strongly mitigated while the intra-molecular dehydration of glycerol is enhanced. These catalytic results suggest that the increased selectivity and activity for the production of acrolein by the addition of sulfolane are ascribed to the dispersing effect of sulfolane on glycerol and to the promoting effect of sulfolane on the diffusion of glycerol and products, respectively. When the sulfolane/glycerol mass ratio was further increased, the glycerol conversion and the selectivity to acrolein decreased with the concurrent increase in the selectivity to high-boiling-point products. The corresponding reaction rate and the TON for the production of acrolein varied likewise. The reason is unclear. It may be presumably because excess sulfolane impedes the adsorption of glycerol on the active sites of ZSM-5_280 by competitive adsorption, thus lowering the catalytic efficiency. Note that once the actual amount of ZSM-5_280 falls, the glycerol conversion and the selectivity to acrolein fall with the concurrent rise in the selectivity to high-boiling-point products, as shown in Table 2 and Table 3.

At this stage, we have shown the significant improvements in catalytic performance toward the liquid-phase dehydration of glycerol to acrolein over ZSM-5_280 using sulfolane as the dispersing agent. After 5 h of reaction, the acrolein yield and the TON for the production of acrolein reach 46% and 9.2 g_acr_ g_cat_^−1^, respectively. Although the acrolein yield is considerably increased by using sulfolane, it is still inferior to those reported from gas-phase dehydration of glycerol with effective H-ZSM-5 catalysts [18,19,23]. From the reaction data in Figure 1, the initial TOF for the production of acrolein attains 3.2 g_acr_ g_cat_^−1^ h^−1^, which is comparable to those of effective H-ZSM-5 catalysts reported for gas-phase dehydration of glycerol [18,19,23]. In fact, Kim et al. systematically investigated H-ZSM-5-catalyzed gas-phase dehydration of glycerol to acrolein [18]. At 340 °C and atmospheric pressure, their best catalyst H-ZSM-5 (SiO_2_/Al_2_O_3_ = 150 molar ratio) could produce an initial acrolein yield of 54%, corresponding to an initial TOF of 3.9 g_acr_ g_cat_^−1^ h^−1^. Jia et al. studied small-sized H-ZSM-5 catalysts for gas-phase dehydration of glycerol to acrolein [19]. At 320 °C and atmospheric pressure, their H-ZSM-5 (SiO_2_/Al_2_O_3_ = 130 molar ratio) as the best catalyst could result in an optimal acrolein yield of 62% and an optimal TOF of 2.8 g_acr_ g_cat_^−1^ h^−1^. Additionally, Qureshi et al. explored nano H-ZSM-5 catalysts with short channels along the *b*-axis for gas-phase dehydration of glycerol to acrolein [23]. At 320 °C and atmospheric pressure, their synthesized H-ZSM-5 (SiO_2_/Al_2_O_3_ = 150 molar ratio) with short *b*-axis channels as the best catalyst could give rise to an initial acrolein yield of 85%, corresponding to an initial TOF of 3.4 g_acr_ g_cat_^−1^ h^−1^.

The contribution to the improved catalytic performance from sulfolane consists in eliminating or reducing the mass transfer limitations by decreasing the viscosity of the reaction system (i.e., increasing the diffusion of glycerol and products). It is difficult to eliminate the mass transfer limitations at lower sulfolane/glycerol mass ratios. Due to the side effect causing catalyst deactivation that occurs when more amounts of sulfolane are used, it is difficult to assess the mass transfer limitations at higher sulfolane/glycerol mass ratios. However, through the reaction rate measurement, the extent of mass transfer limitations can be roughly estimated. From the reaction data in Figure 1, the initial reaction rate of our ZSM-5_280-catalyzed liquid-phase dehydration of glycerol with a catalyst/sulfolane/glycerol mass ratio of 0.05/3/1 is estimated to be 16 g_gly_ g_cat_^−1^ h^−1^. The initial reaction rates of the reported effective H-ZSM-5-catalyzed gas-phase dehydration of glycerol [18,19,23] are calculated to be 6.8, 4.2 and 3.8 g_gly_ g_cat_^−1^ h^−1^, respectively. From the comparison of the initial reaction rates between our liquid-phase dehydration of glycerol and the reported efficient gas-phase dehydration of glycerol [18,19,23], the obviously superior initial reaction rate of our ZSM-5_280-catalyzed liquid-phase dehydration of glycerol with a catalyst/sulfolane/glycerol mass ratio of 0.05/3/1 suggest that the mass transfer limitations are eliminated or well minimized in our reaction system.

The recycling of the ZSM-5_280 catalyst in liquid-phase dehydration of glycerol to acrolein was studied. At 280 °C and a catalyst/sulfolane/glycerol of 0.05/3/1 mass ratio, the reaction was run for 1 h for each cycle. As shown in Table 5, the glycerol conversion decreased with reaction cycle. The acrolein yield slightly increased and the formation of high-boiling-point products and coke slightly decreased after the second cycle. They decreased and increased, respectively, after the following cycles. The fast catalyst deactivation with the decrease in the acrolein yield and the increase in the formation of high-boiling-point products and coke may be mainly caused by the pore blockage of ZSM-5_280 by the products because the low acidity of ZSM-5_280 (as shown later) may have a minor effect, as assumed previously [14,15].

### 2.2. Comparison of Various Solid Acid Catalysts

After having investigated the influence of the reaction parameters, we screened other solid acid catalysts including H-ZSM-5 (SiO_2_/Al_2_O_3_ = 80 molar ratio), i.e., ZSM-5_80, H_3_PO_4_-modified H-ZSM-5_280 such as 2% P-ZSM-5_280, 10% P-ZSM-5_280, 15% P-ZSM-5_280 and 20% P-ZSM-5_280. H-ZSM-5 with an SiO_2_/Al_2_O_3_ molar ratio of 60–100 is a known catalyst for gas-phase dehydration of glycerol to acrolein [18,19,23]. Supported HPW and supported CsPW have been widely studied in dehydration of glycerol to acrolein [2,5,6], while P-ZSM-5 has yet to be reported in the study of dehydration of glycerol. Table 6 shows the comparative catalytic results of these solid acids in liquid-phase glycerol dehydration at 280 °C and a catalyst/sulfolane/glycerol mass ratio of 0.05/3/1. In general, a high glycerol conversion was obtained over all the catalysts after 5 h of reaction. The acrolein yield and the TON for the production of acrolein followed the order of H-ZSM-5 > P-ZSM-5 > HPW > CsPW. Additionally, the heteropolyacids produced much more coke than the ZSM-5-based catalysts. Between the ZSM-5_280 and ZSM-5_80, the former produced a higher selectivity to acrolein owing to its higher weak-acidity, while the latter led to a higher selectivity to high-boiling-point products due to its higher strong-acidity (as shown later). The addition of H_3_PO_4_ to the ZSM-5_280 catalyst reduced the selectivity to acrolein, and the acrolein yield (or the TON for the production of acrolein) reduced with the concurrent reduction in the selectivity to other low-boiling-point products in favor of an increase in the selectivity to high-boiling-point products. The addition of small amounts of H_3_PO_4_ to ZSM-5_280 (giving 2% P-ZSM-5_280) resulted in a tremendous decrease in the selectivity to acrolein from 46 to 21%, whereas a further addition of H_3_PO_4_ caused a progressive increase in the selectivity to acrolein from 21 to 36%. This phenomenon may be understood as follows: The addition of small amounts of H_3_PO_4_ blocks the acid sites of ZSM-5_280 so that these active acid sites are not accessible to the reactants. Only H_3_PO_4_ acid sites on the zeolite surface are accessible. Thus, with 2% P-ZSM-5_280, the reaction would proceed on the H_3_PO_4_ acid sites rather than on the acid sites of ZSM-5_280. The serious fall in the selectivity to acrolein upon the addition of small amounts of H_3_PO_4_ suggests that the acid sites of ZSM-5_280 are more active than the H_3_PO_4_ acid sites for the production of acrolein. The continuous addition of H_3_PO_4_ definitely increases the total number of H_3_PO_4_ acid sites, thus improving the activity for the production of acrolein compared to that of 2% P-ZSM-5_280. Meanwhile, the selectivity to coke was reduced upon the addition of H_3_PO_4_ (giving 2% P-ZSM-5_280). This implies that the acid sites of ZSM-5_280 are also more active than the H_3_PO_4_ acid sites for the formation of coke. The further addition of H_3_PO_4_ likewise resulted in a rise in the selectivity to coke with increasing acidity. These results suggest that the catalytic properties of the acid sites of ZSM-5_280 and the H_3_PO_4_ acid sites are different for the formation of acrolein, other low-boiling-point products, high-boiling-point products and coke. In contrast, although supported heteropolyacids are more selective than zeolite-based catalysts for the production of acrolein in gas-phase glycerol dehydration [2,5,6], the HPW and CsPW favor the formation of coke via the polymerization of glycerol and products in the liquid-phase glycerol dehydration because of their high strong-acidity (as shown later). Thus, the HPW and CsPW give rise to a lower selectivity to acrolein than the ZSM-5-based catalysts in liquid-phase glycerol dehydration.

By P elemental analysis using inductively coupled plasma–optical emission spectroscopy (ICP-OES), negligible P leaching from 10% P-ZSM-5 was detected after 5 h of reaction, which was consistent with the P content of the spent solid catalyst. In fact, after 5 h of reaction, 100 mL of deionized water was added to the reaction mixture. After stirring for 10 min, 5 mL of the liquid filtrate was collected and handled for ICP-OES analysis. The spent solid catalyst was filtered off and washed with deionized water, ethanol and acetone, followed by drying at 110 °C. As a result, the P content of the spent solid catalyst and the P concentration in the reaction mixture were found to be 9.1% and 4 ppm, respectively, which corresponded to a leaching level of 2.2% P from the solid catalyst. This implies that the ZSM-5-suppored H_3_PO_4_ is significantly responsible for the heterogeneous catalytic results. Therefore, catalytic heterogeneity is basically testified in our reaction system.

### 2.3. Correlation of Catalytic Performance with Catalyst Acid Properties

In light of research on gas-phase dehydration of glycerol, the production of acrolein is associated with the acidity of catalysts [29]. Therefore, it is of importance to correlate the acidity of solid catalysts with the acrolein yield in guiding the catalyst design. Temperature-programmed desorption of ammonia (NH_3_-TPD) is used as an efficient means to study the acidity of solid catalysts. According to previous studies [30,31,32], the acid strengths of zeolite catalysts evaluated by NH_3_-TPD can be classified as weak (<400 °C) and strong (≥400 °C) in terms of temperature of NH_3_ desorption. We adopted this standard to classify the acid strengths of our ZSM-5-based catalysts and heteropolyacids. We expressed the acidity (or acid amount) of the solid catalysts in µmol of NH_3_ desorbed per gram of catalyst. Since NH_3_-TPD profiles often overlap, it is necessary to distinguish the amounts of weak and strong acid sites by curve deconvolution (Appendix A).

Figure 2 shows the comparative NH_3_-TPD profiles of H-ZSM-5 and P-ZSM-5_280. ZSM-5_80 had weak and strong acid sites, producing NH_3_ desorption peaks at 274 and 484 °C, whereas ZSM-5_280 merely had weak acid sites, giving NH_3_ desorption peaks at 86, 181 and 387 °C. From the NH_3_-TPD results by curve deconvolution, ZSM-5_280 possesses a higher weak-acidity, although ZSM-5_80 owes higher strong-acidity and higher total-acidity, as seen in Table 7. The higher weak-acidity may be correlated with the higher selectivity to acrolein over ZSM-5_280, and the higher strong-acidity may be in line with the higher selectivity to other high-boiling-point products over ZSM-5_80. 2% P-ZSM-5_280 also had weak and strong acid sites with NH_3_ desorption peaks at 181, 305 and 432 °C. As the content of added H_3_PO_4_ increased, both the total-acidity and weak-acidity clearly increased. From Figure 2 and Table 7, it is assumed that the interaction of H_3_PO_4_ with different sites inside ZSM-5_280 results in the supported H_3_PO_4_ acid sites with different strengths. The weak-acidity of P-ZSM-5_280 dominates, whereas the strong-acidity of the P-ZSM-5_280 catalyst remains as the minor fraction. As shown in Figure 3 and Figure 4, the weak-acidity or total-acidity of P-ZSM-5_280 increases proportionally with increasing P loading. Meanwhile, with the similar glycerol conversions, the selectivity to acrolein increases proportionally with increasing weak-acidity or total-acidity of P-ZSM-5_280. The relations clearly indicate that the selectivity to acrolein increases proportionally with increasing P loading. However, the selectivity to acrolein does not increase regularly with increasing strong-acidity of P-ZSM-5_280. It is, thus, suggested that the production of acrolein over P-ZSM-5_280 is related to the weak acid sites. At the same time, it is reasonable to assume that the formation of coke increases more or less with increasing total-acidity based on our work and the literature [24]. The increase in the formation of coke would not influence the increase in the acrolein yield over P-ZSM-5_280. Therefore, both the acrolein and coke yields ascend when the total-acidity of P-ZSM-5_280 is increased to 1924 µmol g^−1^, as seen in Table 6 and Table 7.

Figure 5 illustrates the comparative NH_3_-TPD profiles of CsPW and HPW. The NH_3_-TPD profile of CsPW displayed broad, low peak at 106 °C and very broad, weak signal from 250 to 550 °C. The NH_3_-TPD profile of HPW presented sharp, intense peak at 609 °C and shoulder peak at 641 °C with unnoticeable peaks below 400 °C, which are characteristic of NH_3_ desorbed from the strong acid sites of HPW. Evidently, HPW possesses a predominant fraction of strong acid sites, and CsPW contains a great fraction of weak acid sites. The presence of large amounts of strong acid sites may be responsible for the high formation of coke over HPW.

IR spectroscopy of pyridine adsorption is a powerful tool to identify Brønsted and Lewis acid sites [33,34]. Figure 6 shows the subtract IR spectra of pyridine adsorbed on the ZSM-5_280 catalyst in the 1700–1400 cm^−1^ region following the two different treatment processes. When a wafer of ZSM-5_280 was subjected to evacuation at 350 °C followed by pyridine adsorption and evacuation at 200 °C, the resulting subtract spectrum exhibited bands at 1634(w), 1620(sh), 1614(sh), 1597(s), 1582(m), 1543(w), 1495(w), 1487(m), 1447(m) and 1439(sh) cm^−1^, as in Figure 6(a). In virtue of the band assignments of adsorbed pyridine by Parry [33] and Basila et al. [34], the bands at 1634(w), 1620(sh), 1543(w) and 1487(m) cm^−1^ are attributed to pyridine adsorbed on Brønsted acid sites; the bands at 1620(sh), 1582(m), 1495(w) and 1447(m) cm^−1^ are due to pyridine adsorbed on Lewis acid sites; and the bands at 1614(sh), 1597(s), 1487(m) and 1439(sh) cm^−1^ correspond to H-bonded pyridine with surface OH groups. It is known by pyridine adsorption IR studies that H-ZSM-5 and proton form SiO_2_-Al_2_O_3_ contain both Brønsted and Lewis acid sites [23,35,36,37,38]. So is the case with ZSM-5_280. The results with the relative intensity of the 1543(w) and 1447 (m) cm^−1^ bands observed by pyridine adsorption at 200 °C show that the Lewis acid sites dominate over the Brønsted acid sites on ZSM-5_280, which is in accordance with the previous IR study of pyridine adsorption on the proton form SiO_2_-Al_2_O_3_ at 170 °C by Matsunaga et al. [37]. In order to reflect the possible chemical change of catalyst acid sites during the dehydration of glycerol to acrolein at 280 °C with the concomitant generation of water, another wafer of ZSM-5_280 was subjected to exposure to water vapor at 280 °C before pyridine adsorption at 200 °C. It was demonstrated by Xu and coworkers that Lewis acid sites susceptibly convert to Brønsted acid sites in the presence of water on NaY at 20–200 °C and on H-ZSM-5 at 75 °C [39,40,41]. After evacuation at 200 °C, the resulting subtract spectrum consisted of four bands at 1636(s), 1624(s), 1547(s) and 1489(s) cm^−1^, as shown in Figure 6(b). The wavenumbers and relative intensity of these four bands are characteristic of pyridine adsorbed on Brønsted acid sites, according to the band assignments of adsorbed pyridine by Parry [38] and Basila et al. [34]. The absence of bands of pyridine adsorbed on the Lewis acid sites and H-bonded pyridine with surface OH groups implies that the Lewis acid sites fully convert to the Brønsted acid sites in the presence of water vapor at 280 °C, and the pyridine bonded to the surface OH more readily desorbs via evacuation at 200 °C after the ZSM-5_280 catalyst has been hydrated at 280 °C. From our observations of pyridine adsorption on the ZSM-5_280 catalyst, we reasonably deduce that Brønsted acid sites act as the true catalytic species for dehydration of glycerol to acrolein under the reaction conditions, which is in strong support of the possible dehydration pathway from glycerol to acrolein occurring on Brønsted acid sites suggested by Alhanash et al. [22,42]. This contribution provides, for the first time, sound spectroscopic evidence of catalysis for dehydration of glycerol to acrolein by solid Brønsted acid sites.

Based on the catalytic, NH_3_-TPD and pyridine adsorption IR studies, we suggest that Brønsted weak acid sites catalyze dehydration of glycerol to acrolein. For H-ZSM-5, the higher weak-acidity results in the higher selectivity to acrolein. The presence of strong acid sites on ZSM-5_80 does not appear to affect the selectivity to acrolein much. For P-ZSM-5, the relations of the acidity of P-ZSM-5 with the selectivity to acrolein and with the P loading show that the selectivity to acrolein increases proportionally with increasing Brønsted weak acid sites, as H_3_PO_4_ is a Brønsted acid. It is, thus, believed that dehydration of glycerol to acrolein proceeds on the Brønsted weak acid sites of P-ZSM-5. The strong acid sites of P-ZSM-5 give rise to little coke. Herein, we propose overall glycerol conversion paths for liquid-phase dehydration of glycerol with the ZSM-5-based catalysts, as shown in Figure 2. For the heteropolyacids, the very low weak-acidity leads to the low selectivity to acrolein. The strong acid sites seem to cause the formation of coke.

## 3. Materials and Methods

Glycerol (≥99.5%), sulfolane ((CH_2_)_4_SO_2_, 99%), anhydrous acrolein (≥95%), pyridine (99.99%), NH_4_-ZSM-5 (SiO_2_/Al_2_O_3_ = 80–280 molar ratios), HPW (reagent grade) and other chemicals were purchased from commercial sources. The H-ZSM-5 was obtained after 5 h of calcination of NH_4_-ZSM-5 at 500 °C in air. The CsPW and H_3_PO_4_-modified H-ZSM-5 were prepared following reported procedures [43,44]. The details of preparation of the latter are described in the Appendix A. The H-ZSM-5 (SiO_2_/Al_2_O_3_ = 80 molar ratio), H-ZSM-5 (SiO_2_/Al_2_O_3_ = 280 molar ratio) and H_3_PO_4_-modified H-ZSM-5 are denoted as ZSM-5_80, ZSM-5_280 and P-ZSM-5, respectively. The physical properties of the ZSM-5-based materials are shown in Table 8. The H-ZSM-5, P-ZSM-5, HPW and CsPW samples were used as catalysts without further treatment.

The liquid-phase dehydration of glycerol was performed at 200–280 °C under atmospheric pressure N_2_ in a glass flask. The whole reaction apparatus is illustrated in Figure 3. All catalysts were in the powder form. A total of 5–20 g of glycerol was loaded with a catalyst, glycerol and sulfolane at a varying catalyst/glycerol/sulfolane ratio. N_2_ was introduced into the reaction mixture at a flow rate of 120 mL min^−1^ and passed through two cold traps, each of which contained 400 g of deionized water immersed in an ice/water bath. The reaction mixture was stirred at ca. 500 rpm. Due to the inconvenience of the product analytical process, such a batch reaction was allowed to run only for collection and analysis of one set of products was performed (e.g., only for collection and analysis of products after 1 h of reaction at 280 °C). For catalyst recycling, the catalyst was washed inside a flask with 50 mL of sulfolane after the removal of the liquid-phase mixture (for each cycle). After washing, fresh glycerol and sulfolane were added, and the reaction was run under equivalent conditions for the next cycle. After reaction, unreacted glycerol and high-boiling-point products (≥200 °C) were collected from the flask. Acrolein, other low-boiling-point products (<200 °C) and small amounts of unreacted glycerol were collected in the cold traps. The low-boiling-point products included acrolein, glycidol, propionic acid, acetaldehyde, propionaldehyde and 2-propen-1-ol. The high-boiling-point products mostly comprised of glycerol oligomers, such as di-glycerol and tri-glycerol. Glycerol and low-boiling-point products were analyzed by gas chromatography (GC). 2-butanol was used as an internal standard. For the sample preparation of glycerol, 30 mg of 2-butanol and 1 g of water were added to a sample vial with 0.2 g of liquid-phase mixture from the flask. For the sample preparation of low-boiling-point products, 30 mg of 2-butanol was added to a sample vial with 1 g of aqueous solution from each cold trap. The prepared samples were quickly analyzed by GC on a Shimadzu gas chromatograph with an auto-sampler equipped with a capillary column (Phenomenex, ZB WAX, 60 m × 0.25 mm × 0.5 µm) and a flame ionization detector (FID). The temperatures of the injector and the FID were set at 300 and 400 °C, respectively. The chromatography column temperature program was run from 60 to 240 °C at a heating ramp of 10 °C min min^−1^ and kept at 240 °C for 12 min. Glycerol, di-glycerol and tri-glycerol were identified on an Agilent gas chromatograph–mass spectrometer (GC-MS) with a DB-5MS column. The selectivity to high-boiling-point products was approximately estimated by subtracting the selectivity to all detected products and coke from 100%. In the meantime, the mass loss of the reaction mixture in the flask after reaction was measured by weighing. The measured mass loss was consistent with the total amount of low-boiling-point products, sulfolane and glycerol in the cold traps determined by GC, which indicated a good carbon balance. 

The conversion of glycerol, the selectivity to product i, and the yield of product i were calculated as follows:$$X_{\mathrm{glycerol}}\;\left(\%\right)=\frac{\mathrm{mol}\;\mathrm{glycerol}\;\mathrm{fed}-\mathrm{mol}\;\mathrm{glycerol}\;\mathrm{remaining}}{\mathrm{mol}\;\mathrm{glycerol}\;\mathrm{fed}}\times 100$$
$$S_{i\;}\left(\%\right)=\frac{\mathrm{mol}\;i}{\mathrm{mol}\;\mathrm{glycerol}\;\mathrm{fed}-\mathrm{mol}\;\mathrm{glycerol}\;\mathrm{remaining}}\times \frac{Z}{3}\times 100$$
$$Y_{i}\;\left(\%\right)=\frac{\mathrm{mol}\;i}{\mathrm{mol}\;\mathrm{glycerol}\;\mathrm{fed}}\times \frac{Z}{3}\times 100$$
where i and Z stand for product i and the number of carbon atoms of product i, respectively. 

The acidity of the solid catalysts was measured by NH_3_-TPD. The NH_3_-TPD experiments were carried out using a Thermo Scientific TPROD 1100 instrument (manufactured by Thermo Fisher Scientific Inc., Massachusetts, U.S.). A total of 0.10 g of the sample was pretreated in flowing He at 200 °C for 60 min, followed by adsorption of flowing 10% NH_3_ in He at 100 °C for 30 min. After the sample had been flushed with flowing He at 100 °C for 15 min, the TPD was performed from 30 to 850 °C in He at a flow rate of 30 mL min^−1^ with a heating ramp of 10 °C min^−1^. The amount of desorbed NH_3_ was monitored by a thermal conductivity detector. Quantification of desorbed NH_3_ was performed by calibration with the injection of 10% NH_3_ in He pulse at 22 °C. The amount of desorbed NH_3_ was obtained from the integrated peak area relative to the average calibration value from a dozen of injections of the NH_3_ pulse.

The nature of the catalyst active acid sites for dehydration of glycerol to acrolein was determined by IR spectroscopy of pyridine adsorption on a BIO-RAD spectrophotometer (manufactured by Bio-Rad Laboratories, Inc., California, U.S.). The ZSM-5_280 samples were ground and pressed into wafers (40 mg each), and subsequently loaded into an atmosphere-controlled cell. In a regular experiment, such a sample wafer was first subjected to thermal pretreatment at 350 °C under vacuum (6.0 Pa) for 30 min. The spectrum of the pretreated sample wafer was recorded under static vacuum at 22 °C as background. Then, the pretreated sample wafer was exposed to vapor of dried and purified pyridine at 2.1 kPa and 200 °C for 10 min, followed by pumping out under vacuum (6.0 Pa) at 200 °C for 30 min. The subtract spectrum of adsorbed pyridine was recorded under static vacuum at 22 °C against the background spectrum. Since glycerol dehydration is accompanied by the generation of water, another sample wafer in a parallel experiment was first exposed to water vapor at 2.4 kPa and 280 °C for 30 min, considering the presence of in situ generated water influencing the chemistry of catalyst acid sites during the reaction [39,40,41]. Afterward, the sample wafer was treated at 200 °C under vacuum (6.0 Pa) for 30 min to remove excess adsorbed molecular water. The spectrum of the treated sample wafer was recorded under static vacuum at 22 °C as background. Next, the treated sample wafer was exposed to vapor of dried and purified pyridine at 2.1 kPa and 200 °C for 10 min, followed by pumping out at 6.0 Pa and 200 °C for 30 min. The subtract spectrum was recorded under static vacuum at 22 °C against the background spectrum. 

The amount of coke formed during a liquid-phase glycerol dehydration reaction was evaluated by thermogravimetric analysis (TGA) of a spent catalyst sample in flowing air using a TGA Q500 analyzer. A catalyst sample was dried overnight at 110 °C in an oven prior to the TGA. A weight loss profile of ca. 10 mg of the catalyst sample was recorded from 22 to 800 °C with a heating ramp of 10 °C min^−1^. 

The BET surface areas and pore volumes of the ZSM-5-based materials were measured by N_2_ adsorption–desorption on a Quantachrome Autosorb-6B analyzer. The P content of the P-ZSM-5_280 catalyst was determined by ICP-OES on a Varian Vista-MPX CCD with simultaneous ICP–optical emission spectrograph.

## 4. Conclusions

In this work, the H-ZSM-5, P-ZSM-5, HPW and CsPW catalysts were tested in liquid-phase dehydration of glycerol under atmospheric pressure N_2_ in a batch reactor. In order to achieve effective liquid-phase dehydration of glycerol to acrolein, the reaction was conducted using high weak-acidity H-ZSM-5 at high temperatures. Sulfolane could be preferably used as the dispersing agent to enhance the selectivity to acrolein through suppressing the formation of polymers and coke. Having a low viscosity, sulfolane also could promote the diffusion of glycerol and products and, thus, eliminate or minimize the mass transfer limitations. The optimal reaction results were achieved using ZSM-5_280 and sulfolane at 280 °C. The initial TOF for the production of acrolein reached 3.2 g_acr_ g_cat_^−1^ h^−1^, which was comparable to those of the effective H-ZSM-5 catalysts reported for gas-phase dehydration of glycerol [18,19,23].

Brønsted acid sites proved to be responsible for liquid-phase dehydration of glycerol to acrolein. Brønsted weak acid sites favor the selectivity to acrolein. Over the ZSM-5-based catalysts, the selectivity to acrolein increased with increasing weak-acidity. Over the heteropolyacids, the very low weak acid sites led to the low selectivity to acrolein. The strong acid sites of all these catalysts more or less favored the formation of coke.

## Data Availability

Data will be made available upon request.

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
