# Peer review of "Liquid-Phase Dehydration of Glycerol to Acrolein with ZSM-5-Based Catalysts in the Presence of a Dispersing Agent"

_molecules, 2023, doi:10.3390/molecules28083316_

Round 1
Reviewer 1 Report
The manuscript submitted by Wang et al. reports the dehydration of glycerol in liquid phase with sulfolane as solvent using various catalysts. The authors varied the reaction conditions and the catalyst composition / type in order to establish a correlation between the acidity and the selectivity to acrolein and coke. The manuscript is well written, with correct interpretation of the catalytic results and the characterization. Thus. it requires only minor corrections:
- In line 227, the authors state that the difference in conversion results from diffusion limitations. This is not very convincing, since the conversion increases afterwards for a catalyst ratio of 0.05.
- From Figure 1, it seems that the “high boiling point products” are an intermediate in the formation of acrolein, since their selectivity decreases with the reaction time and gives rise to acrolein. What are these “high boiling point products”? An analysis by GC-MS or other techniques would be highly interesting in order to identify the reaction pathway.
Minor remarks:
- The term “inter-molecular dehydration” is highly unusual. Do the authors mean the condensation of glycerol, forming dimers / oligomers?
- Please change pressure units from Torr to Pa or bar.
- Figure 3 is not very nice to read. The authors may split the figure into 2 in order to facilitate the reading.
Reviewer 2 Report
The manuscript “Liquid-Phase Dehydration of Glycerol to Acrolein with ZSM-5-Based Catalysts in the Presence of a Dispersing Agent” is well organized and deals with an interesting topic related to upgrading of glycerol using ZSM-5 catalysts and free Keggin type heteropolyacids.
Why choose homogeneous heteropolyacids (HPAs) to compare with the ZSM-5 zeolite catalysts? The authors used H3PO4 to modify ZSM-5 and thus it seems that this homogeneous acid could allow better comparisons than HPAs. A justification is recommendable. Also, it seems that no catalytic test was performed with H3PO4.
The degree of originality of this work seems diluted, although the results are new. As referred by the authors regarding the existent literature: “Thus far, only a couple of cases of ZSM-5-based catalysts applied in liquid-phase dehydration of glycerol to acrolein have appeared [10,15]. These investigations were not comprehensive. The reaction was conducted either at high pressures (70-180 bar) [10] or with high acid strength ZSM-5 [15]…” and “The activity of the heterpolyacids for the production of acrolein had an order of HSiW > 91 H3PW12O40·14H2O (HPW) > H3PMo12O40·28H2O, revealing that liquid-phase dehydration of glycerol is affected by the acidity and stability of heteropolyacids [16].”.
The P-modified zeolite seems to be a new aspect of this work and could be better studied. Is there leaching of phosphorous from the P-ZSM-5 catalysts? Please check by ICP of at least one of the good performing catalysts. What about regeneration of the catalysts and its reuse? This is important to know.
The Introduction is lengthy and may be shortened. The degree of originality may be made clearer to the reader if the authors include a table that summarizes clearly what was already reported in the literature for ZSM-5 and heteropolyacids and what is new in this work (this may shorten the last paragraph which leaves some doubts as mentioned below).
To clarify differences between this work and literature studies, the authors wrote, in the last paragraph of the Introduction, the following:
1) “What’s more, there has been a lack of investigation on the relationship between the production of acrolein and the catalyst acid properties in the liquid phase.”. This does not apply for ref. 15. The main conclusions of the relationships in ref. 15 should be included and the sentence rephrased.
2) “The reaction was conducted either at high pressures (70-180 bar) [10] or with high acid 114 strength ZSM-5 [15]”. However, the authors did not study the effect of pressure to justify any possible improvements.
3) “ZSM-5-based catalysts applied in liquid-phase dehydration of 112
4) “glycerol to acrolein have appeared [10,15]. These investigations were not comprehensive.” This does not apply for ref. 21 “Influence of reaction parameters on glycerol dehydration over HZSM-5 catalyst”
For the sake of clarity, the differences between the influence of acid properties of this work and those reported in the literature for ZSM-5 should be clarified in the form of a Table specifying (qualitatively) the roles of the types (Lewis, Bronsted) and strength (weak, moderate, strong) of acid sites on the Glycerol conversion, acrolein yield, byproducts and coke. Then, the main similarities or differences between this work in relation to the literature should be specified and clarified.
In the discussion of the catalytic results, it is important to include a detailed table of the reaction conditions (T, t, pressure, glycerol concentration in the feed, catalyst:glycerol mass ratio, dispersing agent; ratio of dispersing agent:glycerol, and the catalytic results of this work (for good performing catalysts) and literatures studies using ZSM-5 and heteropolyacids, such as references 15, 21, etc. Then, the main accomplishments of this work in relation to the literature should be specified and clarified.
May the authors, if possible, include a scheme with the overall reaction of glycerol to acrolein and other low and high boiling point products and coke, and indicate in the arrows of the paths which type of catalyst properties of ZSM-5 is suggested to favor those paths?
Fig.4 may include the spectrum for the non-hydrated zeolite for the reader to better understand the differences.
The discussion of the catalytic results refers sometimes to selectivity without mentioning conversion (e.g., Fig. 3 in the discussion of section 3.3). Selectivity should always be compared at similar conversion. If this is not possible, then one should compare the yield.
The following is confusion: “The selectivity to high-boiling point products was approximately estimated by subtracting the 158 selectivity to all detected products and coke from 100 %.”. Please include separate formulae for the selectivity of high boiling point products and for low boiling point products.
It is important to name the low boiling point products.
Please indicate the approximate temperature value of transition between the low and high boiling point (b.p.) products. This may be the b.p. of the low b.p. product that has the highest b.p.
Please include in the supplementary the mass spectra of diglycerol and triglycerol.
Glycerol dimers, trimer products are very poorly volatile. May the authors please include in the experimental section, the GC temperature program (as well as type and temperature of detector and injector, type of carrier gas) for the sake of reproducibility?
The details of the preparation of the modified ZSM-5 catalysts should be nevertheless included in the supplementary material.
Scientific language: Please revise “ atmospheric N2”, “infra spectroscopy”, “low envelop”.
Please include the values of yield (temperature, time) in the sentence: “The acrolein yield fol-290 lowed the order as H-ZSM-5 > P-ZSM-5 > HPW > CsPW.”
Please include the reaction conditions in the caption of Fig. 3
Round 2
Reviewer 2 Report
In general, the comments for improvements have been considered and the work has been improved.